# A Neurophysiological Pattern as a Precursor of Work-Related Musculoskeletal Disorders Using EEG Combined with EMG

**DOI:** 10.3390/ijerph18042001

**Published:** 2021-02-19

**Authors:** Colince Meli Segning, Hassan Ezzaidi, Rubens A. da Silva, Suzy Ngomo

**Affiliations:** 1Department of Applied Sciences, Université du Québec à Chicoutimi (UQAC), Chicoutimi, QC G7H 2B1, Canada; colince.meli-segning1@uqac.ca (C.M.S.); Hassan_Ezzaidi@uqac.ca (H.E.); 2Laboratoire de Recherche Biomécanique et Neurophysiologique en Réadaptation Neuro-Musculo-Squelettique (*Lab BioNR*), Université du Québec à Chicoutimi (UQAC), Chicoutimi, QC G7H 2B1, Canada; Rubens_DaSilva@uqac.ca; 3Department of Health Sciences, Université du Québec à Chicoutimi (UQAC), Chicoutimi, QC G7H 2B1, Canada

**Keywords:** EEG, β.TRPI, EMG, pain, musculoskeletal disorders, manual task

## Abstract

We aimed to determine the neurophysiological pattern that is associated with the development of musculoskeletal pain that is induced by biomechanical constraints. Twelve (12) young healthy volunteers (two females) performed two experimental realistic manual tasks for 30 min each: (1) with the high risk of musculoskeletal pain development and (2) with low risk for pain development. During the tasks, synchronized electroencephalographic (EEG) and electromyography (EMG) signals data were collected, as well as pain scores. Subsequently, two main variables were computed from neurophysiological signals: (1) cortical inhibition as Task-Related Power Increase (TRPI) in beta EEG frequency band (β.TRPI) and (2) muscle variability as Coefficient of Variation (CoV) from EMG signals. A strong effect size was observed for pain measurement under the high risk condition during the last 5 min of the task execution; with muscle fatigue, because the CoV has decreased below 18%. An increase in cortical inhibition (β.TRPI >50%) was observed after the 5th min of the task in both experimental conditions. These results suggest the following neurophysiological pattern—β.TRPI ≥ 50% and CoV ≤ 18%—as a possible indicator to monitor the development of musculoskeletal pain in the shoulder in the context of repeated and prolonged exposure to manual tasks.

## 1. Introduction

Musculoskeletal disorders (MSDs) are “a set of symptoms and inflammatory or degenerative lesions of the musculoskeletal system” associated with the neck, back, and upper and lower extremities of the body [1]. They result from a combination of damage that is caused by exceeding the capacity to adapt and repair structures and they are expressed by pain and other symptoms. such as muscle weakness, stiffness, and a reduced range of motion. Ergonomic factors in the workplace are well-known to be sources of overuse of anatomical tissues when engendering repetitive movements, high cadences, or constrained postures [1]. Considerable efforts and strategies were made in ergonomic adaptation and optimization of workstations for MSDs prevention, such as the rotation of workstations to break repetitive motion cycles, or even varying anatomical structure loading. Despite these efforts, MSDs prevalence remains a major health concern and global burden [2]. Manual work has already been associated with constrained conditions generating disorders at the muscular level. In Nordander et al. (2009)’s study, for repetitive and constrained work, one woman out of two, and one man out of three reported complaints in the neck/shoulder area in the last seven days [3]. A repetitive movement or sustained activity promotes increased muscle fatigue [4] that could be evaluated while using surface electromyography (EMG) estimates [5,6,7,8,9].

Moreover, it is now well documented that, beyond the biomechanical constraints related to manual tasks, appropriate brain responsiveness is required to ensure optimal and safe performance [10,11]. This behavior can be seen as a deficit in neuromuscular stabilization for motor coordination and/or execution of movement [12], exposing anatomical tissues to subtle, inconsistent movements, and, therefore, to micro-injuries from a work-task performance. Finally, repetitive exposure to micro-injuries could increase the risk of developing MSDs and pain and/or other related symptoms. In addition to biomechanical studies regarding motor performance and ergonomic constraints, the development of MSDs should also be further studied from the perspective of motor control. In particular, the study of cerebral behavior via the analysis of electroencephalographic (EEG) signals could help to better understand the production of MSDs in motor performance, while EMG measures could represent peripherical physiological information, such as muscle fatigue as a predictor of MSDs in the workplace. Indeed, motor performance involves neural processes that require cortical inhibition and/or cortical facilitation [13]. Using EEG signal measurements, cortical facilitation can be assessed as Task-Related Power Decrease (TRPD); that is, an increase in neuron excitability; while an Task-related Power Increase (TRPI) may represent cortical inhibition [13,14,15]. The TRPD/TRPI patterns reflect sensorimotor activation/deactivation balance. Therefore, it is possible to use EEG and EMG simultaneously in the physiological assessment for determining the neurophysiological signals pattern that is associated with the development of musculoskeletal pain induced by biomechanical constraints. However, a more biomechanics and neurophysiological study is warranted. In fact, it would be interesting for the first time to know whether central and peripheral neurophysiological signals behavior could be mediated by a realistic manual work tasks, such as an experimental hairdressing workstation simulating a high risk of MSDs development.

Thus, this study aimed to determine the neurophysiological signals responses during hairdressing workstation as a possible indicator to monitor the development of musculoskeletal pain in the shoulder in context of repeated and prolonged exposure to manual tasks. As several studies reported TRPI in relation with pain induced by repetitive task [15,16], we hypothesized that specific cortical and peripheral behavior would be observed under ergonomically high-risk hairdressing workstation condition, which, in turn, could be associated to pain and MSDs development.

## 2. Materials and Methods

### 2.1. Participants

Twelve (12) young healthy volunteers (10 males and two females, all right-handed) participated in this study. Their mean age was 27.6 years (range: 23–35). Pain or discomfort in body sites were assessed with the French version of the Nordic Musculoskeletal Questionnaire (NMQ) [17,18]. None of the participants was a hairdresser (the realistic task chosen) and they had no history of pain, neurological or psychiatric disorders, and neck or shoulder surgeries in the seven days prior to the study. All of the participants gave written consent for their participation. The local Research Ethics Committee (CER #602 558 01) approved the study.

Table 1 illustrates descriptive statistics for demographic (age) and anthropometric variables from all participants, while we evaluated 10 men and two women for a total of 12 participants.

### 2.2. Experimental Conditions

An experimental hairdressing workstation was set up in the laboratory with two experimental condition tasks, i.e., one condition with a high risk of MSDs development (namely, HR condition) and another with a low risk (namely, LR condition). The hairdresser task consisted in a hairdressing dummy head fixed to a telescopic table, which was adjustable to each individual’s height. In the HR condition, the dummy was set at the height of the hands in a constrained position for the arms, i.e., with elbows flexed at 60 degrees in order to constrain working with arms above shoulder level. In the LR condition, arms were in a comfortable posture, i.e., with elbows flexed at 90 degrees. Elbow angles were adjusted in a static position with a manual goniometer. Each task was executed during 30 min in a standing position inside a 1 m^2^ perimeter. It has been determined that thirty (30) consecutive minutes in the same posture is detrimental to musculoskeletal health [19,20,21]. In both conditions, repetitive manual gestures were set with the intention of creating either discomfort/fatigue or pain, among others, in the deltoid muscle. The combination of the prolonged standing posture in a 1 m^2^ perimeter with repetitive movements and arms above shoulder level exposed to MSDs pain or other related symptoms [22]. Even more in a HR condition; and, subsequently exposed anatomical tissues to micro-injuries.

It is known that work-related MSDs occur with task operation execution time cycles of 30 s/cycle or less [23,24]. In the HR condition task, manual operations were executed at a cadence of 30 s/cycle; a metronome beat helped the participants to maintain this cadence. One cycle consisted in the following manual operations: taking a comb in the left hand and scissors in the right hand; cutting locks of hair by small pieces, from the tip to the root. In the LR condition, the cadence was set at 60 s/cycle. The two conditions were separated by a 15-min period of complete rest. The conditions were randomly executed by participants. Each participant had to practice the LR version of the task for 5 min, to ensure that that they fully understood the instructions.

### 2.3. Data Collection

MSDs were assessed with the French version of the Nordic Musculoskeletal Questionnaire (NMQ) [17,18]. The NMQ is a self-administrated standardized questionnaire collecting sociodemographic information, health status, and MSDs body sites. The NMQ is repeatable, sensitive, and useful as a screening tool for musculoskeletal disorders [17]. We used it once, before starting the experiment.

During each experimental conditions, pain intensity was assessed with a verbal numerical rating scale (NRS), where 0 corresponds to no pain and 10 to the worst imaginable pain [25]. Pain scores were collected at baseline and every 5 min. Subsequently, EEG and EMG data were simultaneously collected at baseline and during the task performance (see following sections for measurement details).

#### 2.3.1. EEG Measurement and Processing

EEG signals were recorded using an Emotiv EPOC^®^ EEG Headset (Emotiv Systems Inc., San Francisco, CA, USA) consisting in a wireless, affordable, and reliable device with 14 electrode positions in a 10/20 system [26]. Emotiv EEG provides real time control of electrode contact quality. Impedance was maintained in a 10–20 kΩ range while using saline liquid. The internal sampling rate of the device is 2048 Hz. The data were then digitalized using the embedded 16-bit ADC with 128 Hz sampling frequency per channel before being wirelessly transmitted to the computer. The digitized EEG signals were filtered using a passband filter (0.16 Hz–43 Hz) and 5th-order sinc notch digital filter at 60 Hz (for North America) to eliminate the power line.

First, at baseline, the EEG was collected when the participant was at rest, standing in a neutral position at a distance of 1.5 m facing a round 2 cm-diameter black spot that was drawn on a white wall. The baseline measures were taken during (1) 30 s with eyes being closed and (2) 30 s with eyes open and fixed on the black spot. These baseline measures were used for the normalization of subsequent EEG data. Subsequently, a total of 30 min of EEG signal for each experimental condition was recorded (both LR and HR).

Data EEG preprocessing (illustration in Figure 1) was conducted using MATLAB 2016 to quantify cortical activity using TRPD/TRPI. The first step was the suppression of the DC offset. The EEG signal is stored as floating-point values directly converted from the unsigned 16-bit ADC output from the Emotiv headset. Emotiv DC level occurs at approximatively 4200 µV. During data transfer from the headset to the computer, negative data are those that are less than 4200 µV, and the positive are higher than 4200 µV. We used the simplest method, which consisted in subtracting the average value (approximately 4200 µV) from the entire data of the selected channel. The second step was the selection of the band of interest. The beta EEG frequency band (β.EEG) was selected for analysis in the present study, according to the previous studies [27,28]. β.EEG frequency oscillates from 13 Hz to 30 Hz; we applied the fifth-order Butterworth band pass filter (13 Hz–30 Hz) [29]. Within the beta EEG band, artifacts due to eye blinks or eye movements and EMG were removed using the independent components analysis (ICA) method [30,31], which decomposes the EEG signal into independent components (ICs), allowing for the removal of artifact components. ICA is the most largely used method for removing EEG artifacts [32]. Each EEG recording was decomposed into 17 components using an fast-independent component analysis (Fast-ICA) algorithm, as previously suggested by Molina et al. [33]. Components that were related to eye-blinks and muscle artifacts were discarded after visual inspection. In order to select the artifactual components, we used criteria based on the dominant frequency of the ICA component (i.e., low frequency for eye-blinks, high frequency for muscle artifacts). Spurious spike artifacts from cardiac and respiratory origin remain difficult to eliminate using the ICA method. Therefore, an additional outlier replacement filter was used to remove it [34]. This filter finds outliers in EEG data and replaces them according to a selected fill method. In this study, fill method using linear interpolation of neighboring non-outlier values was used. Finally, we proceeded to normalization of the signal. We applied the zero-mean normalization (z-score) method on all artifact-free EEG signals, during baseline and both LR and HR conditions. This method of normalization was selected among the others because of its highest accuracy [35]. Thus, the normalization equation is presented, as follows:(1)$$\beta .{\mathrm{EEG}}_{N}=\;\frac{\beta .\mathrm{EEG}-\mu \;}{\sigma },$$
where μ represents mean of the EEG signal, σ the standard deviation, β.EEG the total signal, and β.EEG_N_ the normalized EEG signal. The EEG data were normalized, because the absolute value of the EEG signal can vary widely with age, between participants according to part of head recordings, etc. [36]. Therefore, normalization is an essential step to have the same dominator for comparing the changes in the EEG signals from one condition to another.The Power Spectral Density (PSD) in the beta EEG frequency band was computed as a correlate of the degree of beta activity of the targeted neuronal population [37]. When the PSD of the signal is high, it means that there is an increase in the neuronal activity. On the contrary, a decrease in neuronal activity is translated by a decrease in PSD. One must first calculate the autocorrelation function of the signal, followed by the application of the Fast Fourier Transform (FFT) on that autocorrelation function in order to obtain the PSD of a non-stationary signal such as an EEG signal [38]. Autocorrelation function ($r_{\mathrm{xx}}\left(m\right))$ of the normalized beta EEG signal ($\beta .{\mathrm{EEG}}_{N}$).
(2)$$r_{\mathrm{xx}}\left(m\right)\;=\frac{1}{N}\sum_{n=0}^{N-\left|m\right|-1}x\left(n\right)x\left(n+m\right),$$
where $x\left(n\right)$ is the $\beta .{\mathrm{EEG}}_{N}$ signal and N the length of $x\left(n\right)$.PSD estimation from normalized EEG signal ($\beta .{\mathrm{EEG}}_{N}$).

The PSD equation is:(3)$$\mathrm{PDS}\left\{x\left(n\right)\right\}=\;\sum_{m=-\left(N-1\right)}^{\left(N-1\right)}r_{\mathrm{xx}}\left(m\right)\exp\left(-j\omega m\right),$$

ω the angular frequency of signal in radian/s.

The PSD of $\;\beta .{\mathrm{EEG}}_{N}$ signal was computed in all conditions: Baseline (with eyes closed and open), LR, and HR conditions. After PSD was obtained, Task-Related Spectral Perturbation (TRSP) was computed within a 2-s sliding window, with an overlap of 50% in the resting and task conditions, respectively. Resting condition TRSP represents the variation of PSD at rest with eyes open relative to the reference (PSD at rest with eyes closed) in each sliding window; while TRSP during the task (LR/HR conditions) is the change in PSD relative to the reference (PSD at rest with eyes open) in each sliding window [39].
(4)$$\beta .\mathrm{TRSP}\left(k\right)=\;\frac{A^{\beta }\left(k\right)-{\;R}^{\beta }}{R^{\beta }},$$
where $A^{\beta }$ are PSD at rest with eyes open (for β.TRSP in the resting condition) and during the LR and HR experimental conditions (for β.TRSP in task conditions). k represents the samples (128) in one sliding window. $R^{\beta }$ represents the average of PSD in the baseline condition with eyes closed (for β.TRSP in the resting condition) and open (for β.TRSP in tasks conditions), respectively. TRPD and TRPI correspond to the negative and positive value of TRSP, respectively. Therefore, β.TRPI and β.TRPD are expressed in percentage and are defined, respectively, as the ratio of the number of positive and negative TRSP values on total TRSP values, multiplied by 100. The following equation of β.TRPD/TRPI gives the other neurophysiological measurement of the present study, including PSD and CoV. One must remember that the TRPD/TRPI pattern reflects sensorimotor activation and deactivation [13].
(5)$$\beta .\mathrm{TRPD}/\mathrm{TRPI}=\;\frac{N_{\beta .\mathrm{TRSP}}/P_{\beta .\mathrm{TRSP}}}{T_{\beta .\mathrm{TRSP}}}\times 100,$$
where $N_{\beta .\mathrm{TRSP}}$, $P_{\beta .\mathrm{TRSP}}$, and $T_{\beta .\mathrm{TRSP}}$ represent the number of negative, positive, and total β.TRSP values.

Because all of the participants were right-handed, and right body movements are mainly controlled by the left cerebral hemisphere, data recorded on the left fronto-central electrode (FC5) were retained for analysis. FC5 covers the left motor cortex, which controls the movement in the right upper limb. A higher β.TRPI corresponds to more cortical inhibition.

#### 2.3.2. EMG Measurement and Processing

The EMG signals were collected with a wireless FREEEMG 300 (BTS Bioengineering^®^, Milan, Italy). After skin preparation, the EMG electrodes were placed bilaterally on the anterior deltoid muscle. The deltoid muscle largely contributes to the biomechanics of the shoulder joint [40], and during movements, such as those seen in hairdressing. Before starting the experiment, three maximal voluntary isometric contractions (MVC) were obtained for EMG normalization purposes [41]. A 20-s rest period was respected between each MVC. The EMG signals were sampled at 2000 Hz, amplified (×1000), and then filtered through a third-order Butterworth bandpass filter at 25 and 450 Hz. Because of the EMG device’s technical limitations, a continuous 30 min recording was not possible as with an EEG device. The EMG signal was recording during the first 5 min (t_0_–t_5_) at the beginning of the task, then from t_10_ to t_15_, and at the end of task from t_25_ to t_30_ were subsequently computed for both HR and LR conditions.

Figure 2 illustrates data preprocessing and quantification of variability of muscle activity. The choice of the time window can directly affect the performance of the power spectral density (PSD) variation. The size of time window determines the temporal resolution. The selection of an appropriate time window size is vital in order to capture the local muscle dynamic. The relatively short time window can attempt to track spectral variation with time, such as PSD, but it does not adopt an optimal time or frequency resolution for the nonstationary signal [42]. In this study, after filtering, EMG signals were rectified using a RMS method over a 100 ms window according to the previous study [43].

The following equation presents the RMS estimation, i.e., the square root of the average power of the EMG signal for a given period:(6)$$\mathrm{RMS}=\sqrt{\left(1/N\right)\sum_{k=1}^{N}{\left[x_{k}\right]}^{2}},\;k=1,\;2,\;\ldots ,\;N,$$
where N is the number of samples and $x_{k}$ the k-sample of free artifacts EMG data.

The signals were then normalized by MVC [44], as follows:(7)$${\mathrm{EMG}}_{N}\;\left(\%\right)=\;\frac{\mathrm{EMG}\;\mathrm{RMS}\left({\mathrm{task}}_{\mathrm{LR}/\mathrm{HR}}\right)}{\;\mathrm{EMG}\;\mathrm{peak}\;\mathrm{RMS}\left(\mathrm{MVC}\right)\;}\;\times 100,$$

Because the EMG signal is originally recorded in the time domain, the Fourier transform was used to transpose the normalized EMG $\left({\mathrm{EMG}}_{N}\right)$ signals into the frequency domain. Subsequently, the level of neuronal activity in the muscles was quantified by computing the PSD form ${\mathrm{EMG}}_{N}$ signals during the task [5,6,7,8,9]. The following equation presents the PSD estimation [45]:(8)$$\mathrm{PSD}=\frac{1}{T}{\left|\sum_{t=1}^{T}{\mathrm{EMG}}_{N}\left(t\right)\exp\left(-j2\pi \mathrm{ft}\right)\right|}^{2},$$
where T is the total number of measurements. The lower the PSD, the more the muscle is subjected to fatigue; the higher the risk of incoherent movements, the greater the exposure to micro-injuries. We consider PSD to be an indicator of the development of pain or other symptoms of work-related musculoskeletal disorders.

The Coefficient of Variation (CoV) is another EMG indicator to estimate muscle variability. The lower the CoV, the lower the muscle variability, the more the muscle is exposed to fatigue.
(9)$$\mathrm{CoV}=\frac{\mathrm{SD}}{\mathrm{Mean}}\;\times 100,$$
where SD and Mean represent the deviation standard and mean of PSD [46].

Several studies have shown that high muscle variability can have a protective effect against the onset of muscle fatigue and, therefore, a protective effect on the development of MSDs [47,48,49,50].

### 2.4. Statistical Analysis

Data normality was confirmed using the Shapiro–Wilk Test. The mean and standard deviation values were used for descriptive analysis. Neurophysiological variables (PSD_EMG_, PSD_EEG_; CoV, EMG, TRPD/TRPI) were divided into three-time segments for statistics: Baseline; 1 min < t ≤ 5 min; and 5 min < t ≤ 30 min. A repeated ANOVA was used to compared two conditions (HR and LR) across times (1 min < t ≤ 5 min and 5 min < t ≤ 30 min) of pain measure and neurophysiological variables (PSD_EMG_, PSD_EEG_; CoV, EMG, TRPD/TRPI). When necessary, a Post Hoc test was used to localize the differences across times. In addition, the Effect Size (ES) and %Δ from mean difference conditions and times were calculated. The ES was calculated based on the Cohen criteria, i.e., *d* = 0.2 to 0.49 is small, *d* = 0.5 to 0.79 is medium, and *d* ≥ 0.8 is large [51]. Finally, the Pearson’s correlation coefficients were used to determine the relationship between neurophysiological variables and pain measurements across experimental conditions. Significance level was set at *p* < 0.05 and all of the statistical analyses were conducted with SPSS version 24 (IBM Corp., Armonk, NY, USA).

## 3. Results

Table 2 illustrates the descriptive statistics for both experimental conditions of pain scores and neurophysiological variables, i.e., Power Spectral Density (PSD), Coefficient of Variation (CoV), and Task-Related Power Increase/Deacrease in the beta EEG frequency band (β.TRPI/TRPD).

The results from Table 2 denote: (1) a significant increase in pain intensity in the HR condition during the first 5 min of the task (*p* < 0.05); and, on the rest of the task (*p* < 0.01). (2) No significant difference of muscle activity was found across times. (3) Cortical inhibition evolved similarly in both conditions: A significant decrease in the first 5 min (*p* < 0.001); then, a significant intensification (*p* < 0.001) for the time remaining on the task relative to the baseline.

### 3.1. Time and Experimental Conditions Factor on the Pain Score

Figure 3 illustrates the comparison of pain score between the experimental task conditions (LR vs. HR) across times.

A strong effect size between the experimental task conditions was also observed (Es = 1.6 and 2.3 at time points t_10_ and t_30_, respectively), indicating the strong impact of high biomechanical constraints on pain production.

No significant interaction was found between condition X time for the pain variable (*p* > 0.05). However, significant differences were reported for each factor: condition and time. First, high risk condition (HR) reported a significant (*p* < 0.01) increase in pain from time point 10 (t_10_) to the end of task (t_30_).

Both of the experimental conditions show an increase in pain over time. Post hoc comparisons revealed that the pain score was significantly higher from time point 10 until the end of the tasks (t_10_, t_15_, t_20_, t_25_, and t_30_), as compared to the initial time point t_0_ (baseline).

The chosen tasks and biomechanical constraints aimed at exposing the musculoskeletal system to the development of micro-injuries and, therefore, to the risk of developing pain; these constraints were even more pronounced for the HR condition, as observed in Figure 3**.**

### 3.2. Time and Experimental Conditions Factor on Muscle Activity (PSD)

Figure 4 presents the comparison of PSD between LR and HR conditions across times.

No significant difference was found between both experimental task conditions; however, the percentage of difference between both experimental task conditions was about 14% in the HR condition (Figure 4).

### 3.3. Time and Experimental Conditions Factor on Muscle Variability (CoV)

Figure 5 shows the comparison of CoV between LR and HR conditions across times.

No statistical significance was found (Figure 5), but the results indicate less muscle variability over time in the HR condition (Δ = 25.5%), suggesting a higher sensitivity of measure with phenomenon observed.

### 3.4. Time and Experimental Conditions Factor Cortical Inhibition (β-TRPI)

Figure 6 presents the comparison of β.TRPI between the LR and HR conditions across times.

A significant main effect of time was found for β.TRPI (F (1,44) = 180.6, *p* < 0.001, indicating a higher neuronal synchronization (more cortical inhibition) over time (Figure 6). Average β.TRPI was above 60% after 5 min of task execution, indicating a stronger implication of the inhibitory mechanisms over time, which can be interpreted as a slight increase in brain focus on the tasks over time. The balance (%) between Task-Related Power Increase (TRPI) and Decrease (TRPD) in a given neuronal population constitutes 100% of its activity. Finally, non-significant correlations were found between the main variables of interest.

## 4. Discussion

This study aimed at verifying our hypothesis that a specific neurophysiological pattern appears in relation with MSDs production before clinical statement, in the context of a realistic repetitive manual task. First, a strong effect size was observed for the HR condition during the last 5 min of task execution, since we observed, in our participants who are healthy subjects (with no pain), the onset of pain up to moderate intensity. Muscle performance was found to be weak in the same period, while CoV decreased below 18%. The β.TRPI time-course was similar during both experimental conditions, but with a strong increase after the 5th min until the end of the task (β.TRPI > 50%), suggesting the intensification of the cortical inhibitory process. Our results reveal the following pattern: the activation of cortical inhibition accompanied by muscle fatigue over time. These findings are encouraging for the use of neurophysiological estimates (β.TRPI ≥ 50% and CoV ≤ 18%) to monitor pain and MSDs development under repetitive and prolonged manual task exposition. However, we assumed these results from a first study with this experimental design and analysis model. A more detailed analysis of the time course of changes in CoV and β.TRPI could be more informative regarding the dynamics of brain and muscle activity changes during other occupational task involving other muscle groups.

The exposure to biomechanical risk factors at work, such as repetitive movements, constraints, and prolonged postures, contribute to causing and/or exacerbating MSDs symptoms [52]. Our results revealed a critical time point for pain, which is the 10th min after the beginning of the task (Es = 1.6); suggesting that, from the 10th min of a repetitive task, changes start in cortical excitability. A large size effect on pain by high risk conditions for MSDs development at a workstation was found in the last 5 min of task execution (Es = 2.3) [53]. Pain increased over time, without achieving severe pain (in accordance with the recommendations of our ethics committee), which suggested that the realistic task we proposed was appropriate for this study. The present study supports previous studies, which reported that 30 consecutive minutes of biomechanical constraint is detrimental to musculoskeletal health [19,20,21].

With regard to muscle fatigue in relation to motor performance, as many previous studies, we focused on the measurement of muscle activity using the Power Spectral Density (PSD) of the EMG signal [5,6,7,8,9]. PSD reflects how much muscle varies its activity, namely muscular variability for both mechanical stability and neuromuscular control strategies in response to a signal regarding dynamic joint stability. High variability was likely to be slower to fatigue, and vice versa [54]. For example, Zhang et al. (2011) showed that muscle fatigue sets in over time and that the associated consequences/symptoms are likely to appear at the end of the workday. They studied the muscle activity of the neck-shoulder region during 200 min of sewing machine work in asymptomatic female workers. The results of this study showed that the amplitude of activation of the upper trapezius peaked at the 160th min, which suggested a slowing of its activation for the remainder of the time of the task. In addition, the spectral analysis showed a linear decrease in the EMG signal from the 50th min of the task and moments of fatigue recorded at the 70th, 160th, and 200th min [55]. In contrast, we did not find any change in the power of spectral density. We attribute this difference from Zhang’s result to the different protocols (task, time, muscle, population), particularly to the shorter time of our experiment (30 min), while Zhang found a first change at the 50th min of the task. We believe that PSD is not the most sensitive indicator to express muscle weakness. We tested another variable that also assesses the muscle’s capacity to vary its activity, namely CoV. There are different CoV cutoff scores, depending on the study [56,57]. For instance, Harber et al. reported an average CoV between 13% and 18% under the isometric lifting task condition in participants with back pain [58]. The studies noted that a lower CoV in a dynamic realistic manual task suggested a slight variability in muscle activity, i.e., a fatigable muscle [59], as well as low muscle endurance during repetitive movements [60]. Our finding is in agreement with previous studies, since we found a CoV average value of 17.5% in the HR condition (see Table 2). The CoV seems to show a better sensitivity for the detection of MSDs/pain development [61]. Certain studies also suggest that motor variability components could be task-specific [62], while others suggest that lower muscle variability may be also associated with a lower probability of returning to normal postural strategies, due to the presence of pain [63]. In all cases, the authors agree that lower muscle variability may lead to a higher risk of developing MSDs. However, these specificities may justify the differences in the timing of the onset of micro-injuries and, therefore, the time course of muscle variability. This fact could explain that the PSD was not found to be changed at all in the present study.

With regard to neurophysiological estimates for MSDs, neuroimaging studies suggest that possible disturbances within M1 only arise once chronic pain has developed. However, numerous studies support alterations of M1-cortical excitability during acute pain states [64,65]. Specifically, an acute pain state, from moderate to severe intensity (score > 3), leads to motor cortex excitability impairments [66,67] and, particularly, changes in TRPI, reflecting a change in cortical inhibition [27]. For example, Kaiser et al. used two basic experimental movements (hand and feet) to investigate the effect of the sensorimotor cortex on its activation patterns; they found a stronger desynchronization in the upper β-frequency band [63]. Our findings also show an increase in neuronal desynchronization, but without a significant correlation with the pain score, although it is now well-established that pain has negative effects on movement [68,69,70]. We observed the increase in cortical inhibition for both experimental conditions over time, but being more pronounced in the HR condition from the 10th min of task execution. Our results are consistent with previous studies [70,71]. However, assuming that the specific conditions of a manual task (repetitiveness, duration, rhythm, movement constraints, type of pain, etc.) influence the onset, extent, and evolution of micro-injuries, we have no way of certifying the presence of micro-injuries in this project. Specific conditions are likely to influence the time course of cortical activation [66]; this may explain why the increase in cortical inhibition that we observed is not statistically significant and even why it is not different between the two tasks (LR and HR). This does not mean that this increase of β. TRPI at the 50% threshold is irrelevant from the perspective of the neuromuscular stabilization strategy. No correlation was found between the neurophysiological variables, thus supporting the non-linearity of the neurophysiological phenomena of MSDs production.

Our findings suggest the following neurophysiological pattern—β.TRPI higher than 50% + CoV below 18%—as a possible indicator for monitoring MSDs development under repetitive and prolonged manual task exposure involving the shoulder muscles. The strength of these results is based on the use of EEG and EMG when performing a functional realistic task involving the anterior deltoid muscle. Shoulder injuries represent the third most frequent factor of the development of MSDs, with enormous socioeconomic costs, resulting from a significant effect on the participant’s ability to perform activities of daily living [72]. The prevalence of shoulder pain in the general population is above 67% [73]. In hairdressing, for example, the prevalence of MSDs is a major problem [22], and several studies have pointed out that MSDs can occur as early as the first year in some hairdressers, despite workstation ergonomics [74,75,76]. That is why, among workers in general, and particularly among hairdressers, it is necessary to consider the human factor in preventive measures. Our results support the relevance of the contribution of neurophysiological measurements as upstream indicators of the development or prevention of MSDs. This study takes place at the sub-clinical level, before MSDs set in. We designed the experiment to characterize indicators that elude human perception and self-assessment, as is usually done in occupational health (e.g., using a visual analogue scale for pain assessment). The present study aims at better understanding the contribution of the neurophysiological signal responses in a workstation context, which may justify the differences in MSDs’ development when observing two manual workers at a similar workstation.

Given the increasingly easy access to the market for portable, wireless, reliable, and robust devices that are designed for real work environments, it is now possible to consider the contribution of neurophysiological measurements to target prevention strategies against work-related MSDs for occupational health and safety. The use of the TRPD/TRPI patterns may be a valuable tool for assessing the early development of MSDs.

The main limitation of the present study is the small sample size and the fact that only one muscle was investigated (anterior deltoid). Studies are still needed in order to confirm our results in the perspective of contributing to decision-making in occupational health prevention programs.

## 5. Conclusions

The results of the present study suggest the following neurophysiological pattern—β.TRPI ≥ 50% and CoV ≤ 18%—as a possible indicator for monitoring the development of musculoskeletal pain in the shoulder in the context of repeated and prolonged exposure to manual tasks. This is a new avenue, in addition to ergonomic interventions, for the prevention of MSDs based on the human factor.

## Figures and Tables

**Figure 1 ijerph-18-02001-f001:**
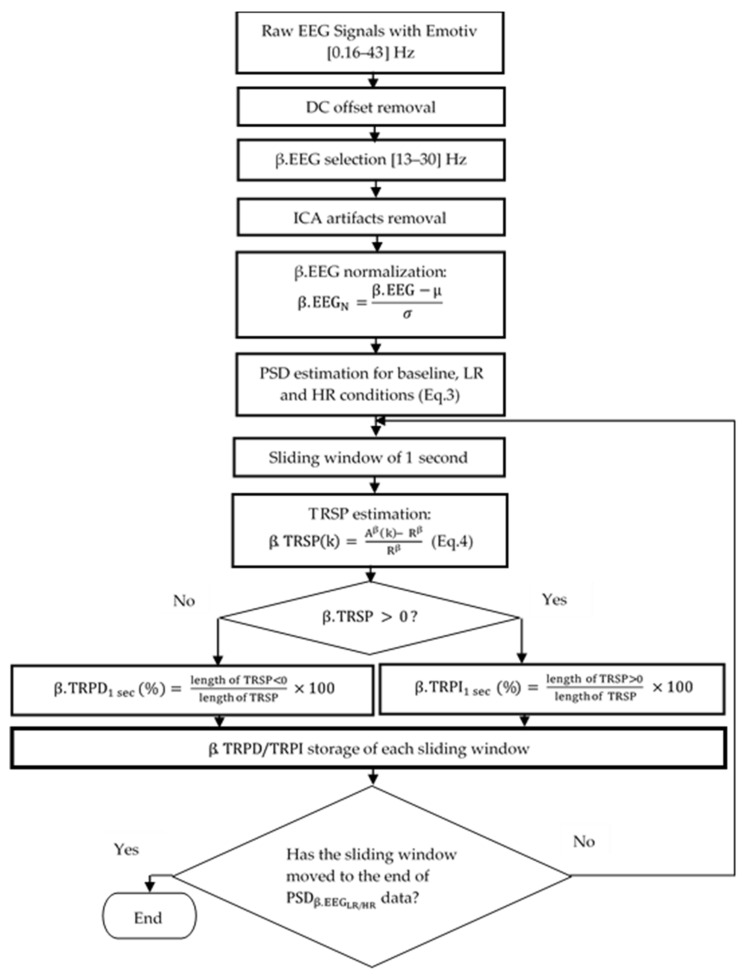
Flowchart diagram of offline electroencephalographic (EEG) preprocessing and Task-Related Power Decrease/Increase in beta frequency band (β.TRPD/TRPI estimation in Matlab software.

**Figure 2 ijerph-18-02001-f002:**
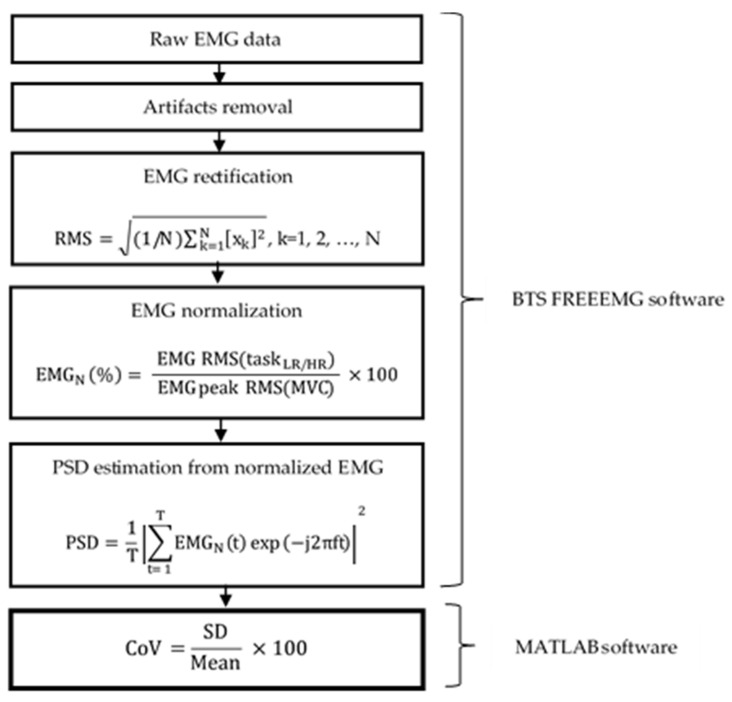
Flowchart diagram of EMG preprocessing and Coefficient of Variation (CoV) estimation.

**Figure 3 ijerph-18-02001-f003:**
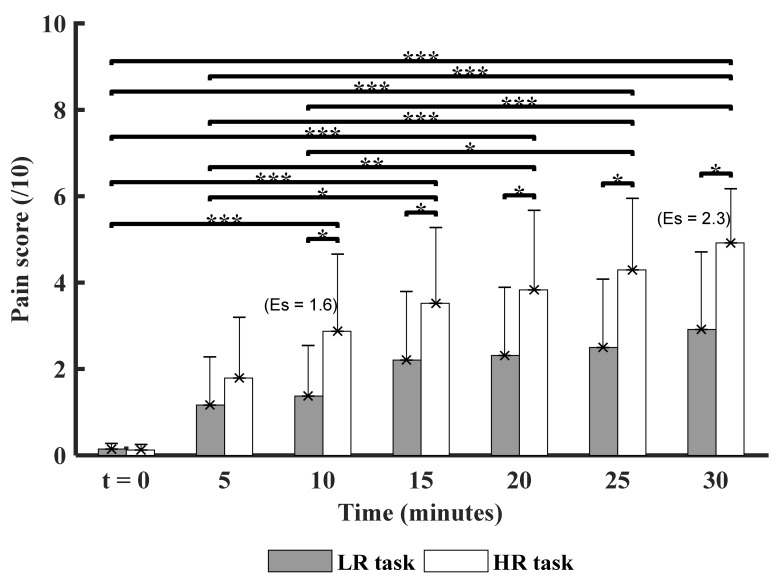
Comparison of mean pain score between both experimental task conditions (LR: Low Risk and HR: High risk) across the 30-min time (t_0_: baseline measure, before starting each task, t_5_, t_10_, t_15_, t_20_, t_25_, and t_30_: at the end of tasks). Significant differences are marked with asterisk: (* *p* < 0.05, ** *p* < 0.01, and *** *p* < 0.001). ES = effect size of conditions on pain intensity.

**Figure 4 ijerph-18-02001-f004:**
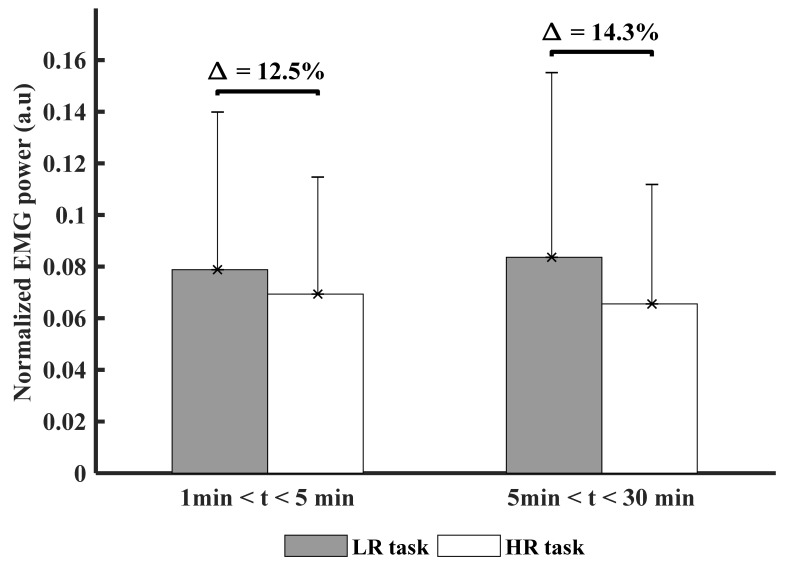
Comparison of the average of the normalized EMG PSD for all participants between both experimental task conditions (LR: Low Risk and HR: High risk). ∆ indicates the percentage of difference between both experimental task conditions.

**Figure 5 ijerph-18-02001-f005:**
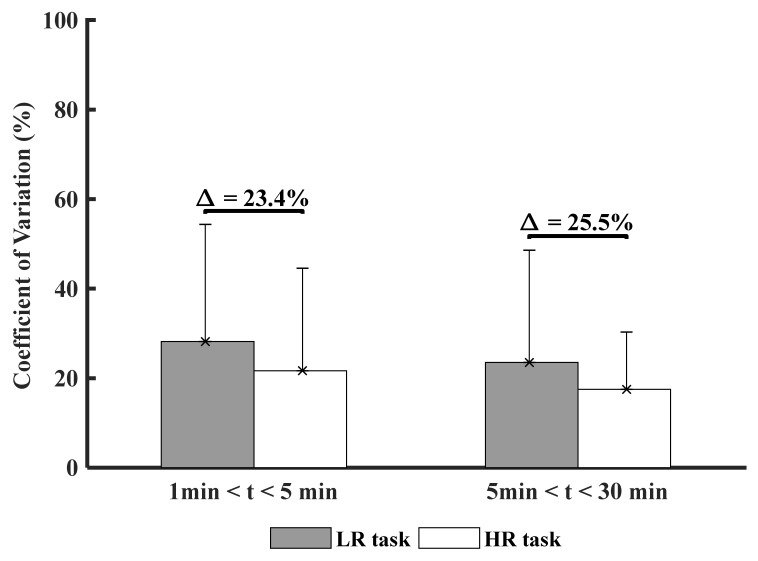
Comparison of the muscle variability for all participants between both experimental task conditions (LR: Low Risk and HR: High risk). ∆ represents the percentage of difference between both experimental task conditions.

**Figure 6 ijerph-18-02001-f006:**
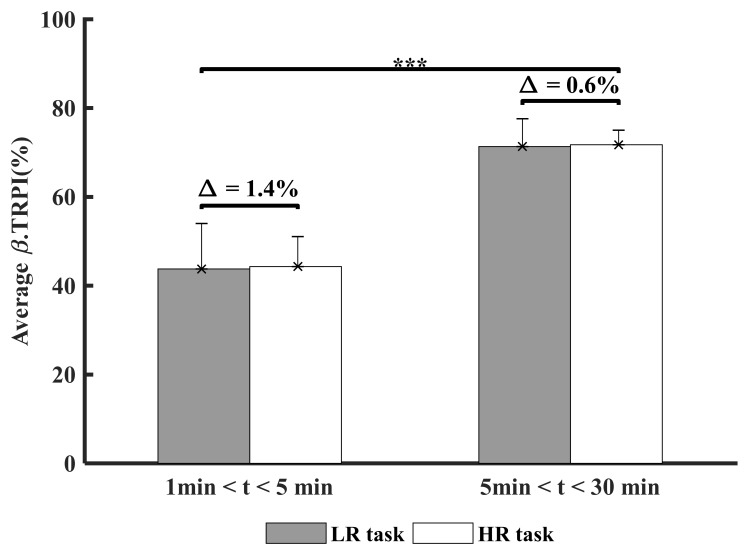
Comparison of the cortical inhibition of all participants between both experimental task conditions (LR: Low Risk and HR: High risk). ∆ indicates the percentage of difference between both experimental task conditions. *** Indicates statistical differences (*p* < 0.001).

**Table 1 ijerph-18-02001-t001:** Participant characteristics.

Characteristics	Men (n = 10)	Women (n = 2)	Mean
Age (years)	26.3 ± 4.3	29.0 ± 2.8	27.6 ± 3.5
Height (cm)	178.2 ± 6.7	165 ± 8.5	171.6 ± 7.6
Weight (kg)	75.8 ± 9.1	69.5 ± 14.8	72.6 ± 11.9
Body mass index	24.0 ± 2.9	25.3 ± 2.9	24.2 ± 2.7

**Table 2 ijerph-18-02001-t002:** Effect of time on pain and neurophysiological variables for high risk (HR) and Low Risk (LR) conditions.

		Baseline		1 min < t ≤ 5 min		5 min < t ≤ 30 min	
		LR	HR	LR	HR	LR	HR
Pain scores (/10)		-	-	1.2 ±1.1	1.8 ± 1.4	2.3 ± 1.5	3.9 ± 1.6
EMG	PSD_EMG_ (V^2^/Hz)	-		0.08 ± 0.06	0.07 ± 0.04	0.07 ± 0.07	0.06 ± 0.05
	CoV_EMG_ (%)	-		28.2 ± 26.2	21.6 ± 22.9	23.5 ± 25.1	17.5 ± 12.8
EEG	β.TRPD (%)	39 ± 1.8		56.2 ± 10.3	55.7 ± 6.7	28.6 ± 6.3	28.3 ± 3.3
	β.TRPI (%)	61 ± 1.8		43.7 ± 10.3	44.3 ± 6.7	71.3 ± 6.3	71.7 ± 3.3

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
