# Peer review of "A Neurophysiological Pattern as a Precursor of Work-Related Musculoskeletal Disorders Using EEG Combined with EMG"

_ijerph, 2021, doi:10.3390/ijerph18042001_

Round 1
Reviewer 1 Report
This study aims to identify neurophysiological markers of musculoskeletal pain caused by repetitive movement and muscle contractions during a simulated occupational task using EEG and EMG. This is an important topic, and the study may contribute to improving our understanding of peripheral and central factors underlying the development of pain syndromes and how to prevent them. The methods are generally adequate and described with appropriate detail. The results are presented with sufficient clarity. Discussion's scope is well defined and pertinent but few statements seem not be well supported by the results as well as the general conclusion. There are also minor issues that should be revised before acceptance for publication.
Abstract
Clear and informative, but the meaning of "neurophysiological precursor pattern" is not readily discernable and the combined use of these terms is not much informative. I suggest the authors try to find a more straightforward way to define the subject of their study.
The EEG variables of interest are named as event-related synchronization (ERS) and event-related desynchronization (ERD). These two terms refer to changes in power in a given EEG band time-locked to a well-defined event. Since there were no events in this study, the use of ERS/ERD terms is not accurate, and other terms should be used, such as task-related band power changes (see Neuper et al., Prog Brain Res, 2006).
Introduction
2nd paragraph, ln. 58, please amend or explain "TMS". (this abbreviation appears repeated times along the manuscript) The 2nd paragraph also offers a brief explanation of the ERS/ERD concepts. This explanation should, however, be made more accurate and should be improved. In particular, ERD is equated with neuronal excitability, wheres ERS is said to indicate neuronal inhibition. This explanation is too simplistic and probably incorrect. In neurophysiology, inhibition and excitation are usually directly related with chages in membrane potential and synaptic activity. Although oscillatory brain activity may also mirror the intrinsic membrane properties and the activity of excitatory and inhibitory local neuronal circuitry, it also reveals the action of larger brain networks and the integrative nature of cortical and subcortical functions. Furthermore, the functional meaning of ERS/ERD is specific for different EEG frequency bands, and this should be explained in the text. I suggest, authors elaborate on the physiological meaning of ERS/ERD and describe specific experimental findings to ilustrate this meaning.
The hypothesis in Introduction is too vague and does not follow any clear evidence or scientific background. What does it mean "a specific neurophysiological pattern"? How does pain relate to ERS/ERD, and what evidence supports such a relationship? These are questions that should be answered along with rewriting the hypothesis.
Methods
EEG processing
Please name the algorithm used for ICA decomposition and the criteria used for selecting the components that were removed from the EMG dataset. Also, provide details about the "outlier replacement filter" used to clean the EEG signals.
Please explain the reason for normalizing the EEG signal since this study follows a repeated measures design with all data collected in a single session.
Page 4, line 160 and following; please indicate that the PSD refers to that in the beta-band, not the whole EEG signal's PSD.
On page 6 we read that the EMG signals were rectified using RMS. Please indicate the time window for computing the RMS and please explain the reason for choosing such time window and how it might have affected the power spectrum density of the EMG signal.
Statistical analysis
Please explain the rationale for selecting the time segments, specifically why comparing a time segment of only 5 minutes with one with 25-minutes duration.
Results
Citations to figures in the text must be edited.
Table 2: please indicate the PSD(EMG) units (arbitrary units, percentual units?). Avoid saying "non statistically significant decrease in muscle activity". If the difference is not statistically significant, we must admit the values are similar or originating from the same statistical distribution.
Page 9, lines. 299-301. No differences in EMG PSD were found between conditions. Still, it is said that higher muscle fatigue occurred in the HR condition. Since the statistical analysis does not support this contention, the authors should avoid such statement.
Discussion
Please explain why CoV = 18% is a cutoff indicating muscle fatigue and the risk of MSD. I can not find any support in the results of the study for such cutoff value. Also betaERS is said to increase after 10 minutes performing the task but results were reported for the entire 5-30 min period.
Page 12, lines. 392-393. When discussing beta-band ERS/ERD, authors say they observe increased desynchronization, but it seems that instead, they found increased beta-band synchronization with task time (see figure 6).
Page 13, lines. 409-410. Considering the lack of correlation between pain scores, CoV, and beta-band PSD, I don't see how this study supports the statement that Cov > 18% and ERS>50% are markers of MSD. A more detailed analysis of the time course of changes in CoV and betaERS could be more informative about the dynamics of brain and muscle activity changes during the occupational task.
Author Response
# Reviewer 1
Overall Comments:
This study aims to identify neurophysiological markers of musculoskeletal pain caused by repetitive movement and muscle contractions during a simulated occupational task using EEG and EMG. This is an important topic, and the study may contribute to improving our understanding of peripheral and central factors underlying the development of pain syndromes and how to prevent them. The methods are generally adequate and described with appropriate detail. The results are presented with sufficient clarity. Discussion's scope is well defined and pertinent but few statements seem not be well supported by the results as well as the general conclusion. There are also minor issues that should be revised before acceptance for publication.
Thank you so much for your time and collaboration to review our manuscript.
Abstract
Comment 1: Clear and informative, but the meaning of "neurophysiological precursor pattern" is not readily discernable and the combined use of these terms is not much informative. I suggest the authors try to find a more straightforward way to define the subject of their study.
ANSWER comment 1: Thank you for this comment. We have removed the term “precursor” to eliminate redundancy and improve information. The sentence is rephrased as follows in lines 13-14:
We aimed to determine the neurophysiological pattern associated with the development of musculoskeletal pain induced by biomechanical constraints.
Comment 2: The EEG variables of interest are named as event-related synchronization (ERS) and event-related desynchronization (ERD). These two terms refer to changes in power in a given EEG band time-locked to a well-defined event. Since there were no events in this study, the use of ERS/ERD terms is not accurate, and other terms should be used, such as task-related band power changes (see Neuper et al., Prog Brain Res, 2006).
ANSWER comment 2: Thank for this comment and we proceeded the change. The reviewer is absolutely right; there was no event in this study, but rather a continue task in two conditions (HR and LR). Indeed, we calculated the overall variance of each condition. Then, in order to estimate the task-related power increase/decrease, each condition was related to the resting condition.
Comment 3: Introduction 2nd paragraph, ln. 58, please amend or explain "TMS" (This abbreviation appears repeated times along the manuscript)
The 2nd paragraph also offers a brief explanation of the ERS/ERD concepts. This explanation should, however, be made more accurate and should be improved. In particular, ERD is equated with neuronal excitability, whereas ERS is said to indicate neuronal inhibition. This explanation is too simplistic and probably incorrect. In neurophysiology, inhibition and excitation are usually directly related with changes in membrane potential and synaptic activity. Although oscillatory brain activity may also mirror the intrinsic membrane properties and the activity of excitatory and inhibitory local neuronal circuitry, it also reveals the action of larger brain networks and the integrative nature of cortical and subcortical functions. Furthermore, the functional meaning of ERS/ERD is specific for different EEG frequency bands, and this should be explained in the text.
I suggest, authors elaborate on the physiological meaning of ERS/ERD and describe specific experimental findings to illustrate this meaning.
ANSWER comment 3:
- TMS: Thank you for that observation. It is the abbreviation of musculoskeletal disorders in French. This is a translation error. We have made the correction to both places, TMS is replaced by MSDs – Line 57 and Line 442.
- ERS/ERD concepts: Even though ERD/ERS and TRPD/TRPI follow the same rationale, i.e., the analysis of regional changes in oscillatory brain activity in relation to a certain event, they differ in that ERD/ERS focuses on spectral power changes time-locked to a single stimulus [1]., whereas TRPD/TRPI concentrates on changes with steady-state processes related to continuous movement execution [2]. As answered in comment 2, we used TRPD/TRPI; hence the non-pertinence to elaborate more here on the concepts of ERS / ERD.
Comment 4 - The hypothesis in Introduction is too vague and does not follow any clear evidence or scientific background. What does it mean "a specific neurophysiological pattern"? How does pain relate to ERS/ERD, and what evidence supports such a relationship? These are questions that should be answered along with rewriting the hypothesis.
ANSWER Comment 4: Thank you for this comment. We reformulated this section in the introduction lines 71-74 as following:
This study aimed thus to determine the neurophysiological signals responses during hairdressing workstation as a possible indicator to monitor the development of musculoskeletal pain in the shoulder in context of repeated and prolonged exposure to manual tasks. As several studies reported TPRD in relation with pain induced by motor task [2, 3] we hypothesized that specific cortical and peripheral behavior would be observed under ergonomically high-risk hairdressing workstation condition, which in turn could be associated to pain and MSDs development.
Methods
EEG processing
Comment 5 Please name the algorithm used for ICA decomposition and the criteria used for selecting the components that were removed from the EMG dataset. Also, provide details about the "outlier replacement filter" used to clean the EEG signals.
ANSWER comment 5: Thank you for this comment.
- Name of algorithm used for ICA: We reformulated this section in the EEG processing lines 171-176 as following:
As previously suggested by Molina et al. [4], each EEG recording was decomposed into 17 components using an fast-independent component analysis (Fast-ICA) algorithm. Components related to eye-blinks and muscle artifacts were discarded after visual inspection. To select the artifactual components, we used criteria based on the dominant frequency of the ICA component (i.e., low frequency for eye-blinks, high frequency for muscle artifacts).
- outlier replacement filter: We reformulated this section about EEG processing lines 178-180as following:
This filter finds outliers in EEG data and replaces them according to a selected fill method. In this study, fill method using linear interpolation of neighboring non-outlier values was used.
Comment 6 Please explain the reason for normalizing the EEG signal since this study follows a repeated measures design with all data collected in a single session.
ANSWER comment 6: Thank you for this comment. We reformulated this section about EEG processing lines 186-190 as following:
The EEG data were normalized because the absolute value of the EEG signal can vary widely with age, between participants, according to the parts of the head recording, etc., [5]. Therefore, normalization is an essential step in order to have the same denominator for comparing the changes in the EEG signal from one condition to another.
Comment 7 Page 4, line 160 and following; please indicate that the PSD refers to that in the beta-band, not the whole EEG signal's PSD.
ANSWER comment 7: Thank you for this comment. We added the term beta in lines 191-192 as following:
The Power Spectral Density (PSD) in the beta EEG frequency band was computed as a correlate of the degree of beta activity of the targeted neuronal population.
Comment 8: On page 6 we read that the EMG signals were rectified using RMS. Please indicate the time window for computing the RMS and please explain the reason for choosing such time window and how it might have affected the power spectrum density of the EMG signal.
ANSWER comment 8: Thank you for this comment. We added information in the EMG processing section; lines 253-255 as following:
After filtering, EMG signals were rectified using RMS method over a 100 ms window; considering that EMG is a non-stationary signal, its analysis is appropriate within a window of 100 ms [5, 6].
Statistical analysis
Comment 9: Please explain the rationale for selecting the time segments, specifically why comparing a time segment of only 5 minutes with one with 25-minutes duration.
ANSWER comment 9: In our experience, it is best to consider physiological changes after 5 minutes of work.
Results
Comment 10: Citations to figures in the text must be edited.
ANSWER comment 10: Thank you for this comment. We edited it, see lines 157, 306, 326, 328, 338, 344,348 and 355.
Comment 11: Table 2: please indicate the PSD (EMG) units (arbitrary units, percentual units?). Avoid saying "non statistically significant decrease in muscle activity". If the difference is not statistically significant, we must admit the values are similar or originating from the same statistical distribution.
ANSWER comment 11: Thank you for this comment.
- PSD(EMG) units: We revised table 2 (line 4) addicting appropriate units as following:
PSDEMG (V2/Hz)
- non statistically significant decrease in muscle activity: We reformulated in line 301 as following: No significant difference of muscle activity was found across times.
Comment 12: Page 9, lines. 299-301. No differences in EMG PSD were found between conditions. Still, it is said that higher muscle fatigue occurred in the HR condition. Since the statistical analysis does not support this contention, the authors should avoid such statement.
ANSWER comment 12: Thank you for this comment. We avoid «muscle fatigue was must pronounced over time under HR condition». The new sentence is as following (334-336):
No significant difference was found between both experimental task conditions. However, the percentage of difference between both experimental task conditions was about 14% in HR condition (Figure 4).
Discussion
Comment 13: - Please explain why CoV = 18% is a cutoff indicating muscle fatigue and the risk of MSD. I can not find any support in the results of the study for such cutoff value. Also, betaERS is said to increase after 10 minutes performing the task but results were reported for the entire 5-30 min period.
ANSWER comment 13: Thank you for this comment.
- Please explain why CoV = 18% is a cut-off indicating muscle fatigue and the risk of MSD:
Apologised, we forgot the term «lower» in our explanation. This is the right sentence (line 413):
Studies noted that a lower CoV in a dynamic realistic manual task suggested a slight variability in muscle activity i.e., a fatigable muscle [6], as well as low muscle endurance during repetitive movements [7].
- betaERS (newly revised as beta TRPI in our study) is said to increase after 10 minutes performing the task but results were reported for the entire 5-30 min period.
We revised in the main manuscript in lines 369-371 as following:
β.TRPI time-course was similar during both experimental conditions, but with a strong increase after the 5th minute until the end of the task (β.TRPI > 50%), suggesting intensification of the cortical inhibitory process.
Comment 14: Page 13, lines. 409-410. Considering the lack of correlation between pain scores, CoV, and beta-band PSD, I don't see how this study supports the statement that Cov > 18% and ERS>50% are markers of MSD. A more detailed analysis of the time course of changes in CoV and beta-ERS could be more informative about the dynamics of brain and muscle activity changes during the occupational task.
ANSWER comment 14: Thank you for this comment. CoV> 18% and TRPI>50% is our proposal based on our interpretation of our findings. That it our main proposal for the neurophysiological signals responses during hairdressing workstation as a possible indicator to the development of musculoskeletal pain in context of repeated and prolonged exposure to manual tasks.
Reviewer 2 Report
Thank you for opportunity to review the paper entitled “A neurophysiological pattern as a precursor of work-related 2 musculoskeletal disorders using EEG combined with EMG”. The aim of the study was to determine the neurophysiological precursor pattern associated with the 13 development of musculoskeletal pain induced by biomechanical constraints. The study was performed on 12 young healthy volunteers. Based on the results obtained from the conducted studies, the authors conclude:” The results of the present study suggest the following neurophysiological pattern - 439 β.ERS ď‚ł 50% and CoV ď‚Ł 18% - as a possible indicator to monitor the development of mus-440 culoskeletal pain in the shoulder in context of repeated and prolonged exposure to manual 441 tasks”.
In my opinion, the paper covers an interesting topic and well written but some elements require additional explanation to readers.
Line 16 and 85-86: How and who assessed whether the manual task is HR or LR?
Line 24, 338, 440: „β.ERS ď‚ł 50% and CoV ď‚Ł 18%” Does it only apply to HR or also to LR?
Line 48-49: The authors write: „However, sometimes the brain can initiate an erroneous motor command, leading to inappropriate muscle contraction patterns”. Why does the brain sometimes initiate an erroneous motor command? In what cases? This needs to be clarified!
Line 58: Abbreviation should be clarified.
Line 80: Why only 7 days?
Line 84-108: It is difficult to imagine the position of the research participants. Pictures of the test bench and the HR and LR positions would be useful.
Line 97-99: On what basis do the authors claim this?
Line 102-105: Why HR 30s / cycle and LR 60s / cycle? Why not the same values?
Line 137, 216, 271, 291, 293, 301, 303, 309, 313, 320, 376: „Error! Reference source not found” Error editing the manuscript or a program error?
Line 204, 413, 435: Lateral or anterior deltoid muscle part?
Line 343-345: Is there any practical conclusion from this?
Line 417-420, 426: What does "human factor", "bio-human factor" mean? Please clarify!
Author Response
# Reviewer 2
Overall Comments:
Thank you for opportunity to review the paper entitled “A neurophysiological pattern as a precursor of work-related 2 musculoskeletal disorders using EEG combined with EMG”. The aim of the study was to determine the neurophysiological precursor pattern associated with the 13 development of musculoskeletal pain induced by biomechanical constraints. The study was performed on 12 young healthy volunteers. Based on the results obtained from the conducted studies, the authors conclude:” The results of the present study suggest the following neurophysiological pattern - β.ERS ³ 50% and CoV £ 18% - as a possible indicator to monitor the development of musculoskeletal pain in the shoulder in context of repeated and prolonged exposure to manual tasks”.
In my opinion, the paper covers an interesting topic and well written but some elements require additional explanation to readers.
Thank you so much for your time and collaboration to review our manuscript.
Comment 1: Line 16 and 85-86: How and who assessed whether the manual task is HR or LR?
ANSWER comment 1: Thank you for this comment. As we mentioned in the experimental conditions section, in lines 119-124 as following:
It is known that work-related MSDs occur with task operation execution time cycles of 30s/cycle or less [8, 9]. In the HR condition task, manual operations were executed at a cadence of 30s/cycle; a metronome beat helped the participants to maintain this cadence. In the LR condition, the cadence was set at 60s/cycle.
Comment 2: Line 24, 338, 440:β.ERS ³ 50% and CoV £ 18%” Does it only apply to HR or also to LR?
ANSWER comment 2: Thank you for this question. This applies to HR, knowing that in this study, HR is very temporary because of ethical constraints. Or, manual workers are really exposed to prolonged ergonomic and biomechanical constraints.
Comment 3: Line 48-49: The authors write: „However, sometimes the brain can initiate an erroneous motor command, leading to inappropriate muscle contraction patterns”. Why does the brain sometimes initiate an erroneous motor command? In what cases? This needs to be clarified!
ANSWER comment 3: Thank you for this comment. In order to avoid any confusion, we excluded the following sentence «However, sometimes the brain can initiate an erroneous motor command, leading to inappropriate muscle contraction patterns».
Comment 4: Line 58: Abbreviation should be clarified.
ANSWER comment 4: Thank you for this comment. We replaced TMS by MSDs in lines 57 and 448.
Comment 5: Line 80: Why only 7 days?
ANSWER comment 5: Thank you for this comment. It was 7 days, because is it well know that memory with regard to pain is not reliable more than 7 days.
Comment 6: Line 84-108: It is difficult to imagine the position of the research participants. Pictures of the test bench and the HR and LR positions would be useful.
ANSWER comment 6: Thank you for this comment. Data collection was carried out in 2018-2019; we have not obtained the participants' permission to publish their images.
Comment 7: Line 97-99: On what basis do the authors claim this?
ANSWER comment 7: Thank you for this comment. In order to clarify our idea, we reformulated the sentence in lines 116-119 as following:
The combination of the prolonged standing posture in a 1m2 perimeter with repetitive movements and arms above shoulder level exposed to MSDs pain or other related symptoms [10], even more in a HR condition; and subsequently exposed anatomical tissues to micro-injuries.
Comment 8: Line 102-105: Why HR 30s / cycle and LR 60s / cycle? Why not the same values?
ANSWER comment 8: Thank you for these questions. As we mentined in lines 119-128, It is known that work-related MSDs occur with task operation execution time cycles of 30s/cycle or less [8, 9]. Thus, 30s / cycle and 60s / cycle were used in order to differentiate both experimental conditions. 30s / cycle is a fast execution of movements, while 60 s/ cycle is slower.
Comment 9: Line 137, 216, 271, 291, 293, 301, 303, 309, 313, 320, 376: „Error! Reference source not found” Error editing the manuscript or a program error?
ANSWER comment 9: Thank you for this comment. We have corrected in lines 157, 306, 326, 328, 338, 344,348 and 355.
Comment 10: Line 204, 413, 435: Lateral or anterior deltoid muscle part?
ANSWER comment 10: Thank you for this comment. We revised in line 241 as following:
After skin preparation, EMG electrodes were placed bilaterally on the anterior deltoid muscle.
Comment 11: Line 343-345: Is there any practical conclusion from this?
ANSWER comment 11: Thank you for this comment. We revised in line 384 as following: suggesting that from the 10th minute of a repetitive task, changes start in cortical excitability.
Comment 12: Line 417-420, 426: What does "human factor", "bio-human factor" mean? Please clarify!
ANSWER comment 12: Thank you for this comment. We suppressed «bio human factors » and we revised the sentence as following in lines 466-467.
The present study aims at better understanding the contribution of the neurophysiological signals responses in workstation context, which may justify the differences in MSDs development when observing two manual workers at a similar workstation.
Reviewer 3 Report
This research is necessary for MSDs prevention, especially combining the EEG and EMG measures. The study is well designed, the writing is good enough, and the results are interesting. The topic is relevant to IJERPH. There are a few concerns mainly addressable through a revision to improve the manuscript.
- Providing the image or illustration of the experimental condition will be better to increase comprehension.
- There is a problem occurred in line 136-137.
- The quality of Figure 1 is not enough. Please rebuild.
- The MVC plays a critical role in calculating the %EMG. Hence, the authors need to address the procedure clearly (e.g., postures, exertion time, repetition…) that was applied in collecting MVC measures. Otherwise, the following data used in the analysis would be troublesome.
- Table 1 can be considered to move to the 2.1 section.
- Ln 254-255, “Descriptive Statistics for demographic (age) and anthropometric variables are illustrated in the Table 1: while men and women were homogeneous for all participants (n = 12).” – What is homogeneous means for? How to ensure it was homogeneous?
- The title of the manuscript should be slightly revised due to the manuscript only investigating one muscle of the shoulder, as the authors mentioned in the limitation.
- Why authors applied hairdressing as the simulation task? Can the results from the hairdressing task be extended to other workstations?
Author Response
# Reviewer 3
Overall Comments:
This research is necessary for MSDs prevention, especially combining the EEG and EMG measures. The study is well designed, the writing is good enough, and the results are interesting. The topic is relevant to IJERPH. There are a few concerns mainly addressable through a revision to improve the manuscript.
Thank you so much for your time and collaboration to review our manuscript.
Comment 1: Providing the image or illustration of the experimental condition will be better to increase comprehension.
ANSWER comment 1: Data collection was carried out in 2018-2019; we have not obtained the participants' permission to publish their images.
Comment 2: There is a problem occurred in line 136-137.
ANSWER comment 2: Thank you for this comment. We added the term Figure 1 in line 147 as following:
To quantify cortical activity using TRPD/TRPI, data EEG preprocessing (illustration in Figure 1) was conducted using MATLAB 2016.
Comment 3: The quality of Figure 1 is not enough. Please rebuild.
ANSWER comment 3: Thank you for this comment. We rebuild the figure 1.
Comment 4: The MVC plays a critical role in calculating the %EMG. Hence, the authors need to address the procedure clearly (e.g., postures, exertion time, repetition…) that was applied in collecting MVC measures. Otherwise, the following data used in the analysis would be troublesome.
ANSWER comment 4: Thank you for this comment. To avoid some much information in the methods, we implied this information; however, the MVC measures were made in accordance with the SENIAM guide.​
Comment 5: Table 1 can be considered to move to the 2.1 section.
ANSWER comment 5: Thank you for this comment. Yes, we changed.
Comment 6: Ln 254-255, “Descriptive Statistics for demographic (age) and anthropometric variables are illustrated in the Table 1: while men and women were homogeneous for all participants (n = 12).” – What is homogeneous means for? How to ensure it was homogeneous?
ANSWER comment 6: Thank you for this comment. We reformulated the sentence in lines 98-101 as following:
Descriptive Statistics for demographic (age) and anthropometric variables from all participants are illustrated in the Table 1, while we evaluated 10 men and 2 women for a total of 12 participants.
Comment 7: The title of the manuscript should be slightly revised due to the manuscript only investigating one muscle of the shoulder, as the authors mentioned in the limitation.
ANSWER comment 7: Thank you for this comment. We assumed a more generic title from short view about the issue investigated. The muscle group here was more arbitrary, but we assumed in the limits of study.
Comment 8: Why authors applied hairdressing as the simulation task? Can the results from the hairdressing task be extended to other workstations?
ANSWER comment 8: Future investigations are necessary.
References
[1] G. Pfurtscheller et F. L. Da Silva, "Event-related EEG/MEG synchronization and desynchronization: basic principles," Clinical neurophysiology, vol. 110, no. 11, pp. 1842-1857, 1999.
[2] C. Neuper, M. Wörtz, et G. Pfurtscheller, "ERD/ERS patterns reflecting sensorimotor activation and deactivation," Progress in brain research, vol. 159, pp. 211-222, 2006.
[3] Y. Hashimoto, J. Ushiba, A. Kimura, M. Liu, et Y. Tomita, "Correlation between EEG-EMG coherence during isometric contraction and its imaginary execution," Acta Neurobiol Exp (Wars), vol. 70, no. 1, pp. 76-85, 2010.
[4] V. Molina et al., "Deficit of entropy modulation of the EEG in schizophrenia associated to cognitive performance and symptoms. A replication study," Schizophrenia research, vol. 195, pp. 334-342, 2018.
[5] Y. Bai, G. Huang, Y. Tu, A. Tan, Y. S. Hung, et Z. Zhang, "Normalization of pain-evoked neural responses using spontaneous EEG improves the performance of EEG-based cross-individual pain prediction," Frontiers in computational neuroscience, vol. 10, p. 31, 2016.
[6] J. Qin, J. H. Lin, B. Buchholz, et X. Xu, "Shoulder muscle fatigue development in young and older female adults during a repetitive manual task," (en eng), Ergonomics, vol. 57, no. 8, pp. 1201-12, 2014.
[7] J. C. Cowley et D. H. Gates, "Influence of remote pain on movement control and muscle endurance during repetitive movements," (en eng), Experimental brain research, vol. 236, no. 8, pp. 2309-2319, Aug 2018.
[8] B. A. Silverstein, L. J. Fine, et T. J. Armstrong, "Hand wrist cumulative trauma disorders in industry," (en eng), British journal of industrial medicine, vol. 43, no. 11, pp. 779-784, 1986.
[9] P. Spielholz, B. Silverstein, M. Morgan, H. Checkoway, et J. Kaufman, "Comparison of self-report, video observation and direct measurement methods for upper extremity musculoskeletal disorder physical risk factors," Ergonomics, vol. 44, no. 6, pp. 588-613, 2001.
[10] A. Kozak, T. Wirth, M. Verhamme, et A. Nienhaus, "Musculoskeletal health, work-related risk factors and preventive measures in hairdressing: a scoping review," Journal of Occupational Medicine and Toxicology, vol. 14, no. 1, pp. 1-14, 2019.